# Efficacy of Denosumab Therapy Following Treatment with Bisphosphonates in Women with Osteoporosis: A Cohort Study

**DOI:** 10.3390/ijerph18041728

**Published:** 2021-02-10

**Authors:** Chiara Marocco, Giovanna Zimatore, Edoardo Mocini, Rachele Fornari, Giovanni Iolascon, Maria Chiara Gallotta, Viviana Maria Bimonte, Carlo Baldari, Andrea Lenzi, Silvia Migliaccio

**Affiliations:** 1Department of Movement, Human and Health Sciences, Health Sciences Section, University Foro Italico of Rome, 00135 Rome, Italy; chiara.marocco@hotmail.it (C.M.); viviana.bimo@gmail.com (V.M.B.); 2Department of Theoretical and Applied Sciences, eCampus University, 22060 Novedrate, Italy; carlo.baldari@uniecampus.it; 3IMM-CNR, Institute for Microelectronics and Microsystems, 40129 Bologna, Italy; 4Department of Experimental Medicine, Medical Pathophysiology, Endocrinology and Nutrition Section, University Sapienza of Rome, 00185 Rome, Italy; emocini@gmail.com (E.M.); fornari.rachele@gmail.com (R.F.); andrea.lenzi@uniroma1.it (A.L.); 5Department of Medical and Surgical Specialties and Dentistry, University of Campania “Luigi Vanvitelli”, 81100 Caserta, Italy; giovanni.iolascon@unicampania.it; 6Department of Physiology and Pharmacology “Vittorio Erspamer”, Sapienza University of Rome, 00185 Rome, Italy; mariachiara.gallotta@uniroma1.it

**Keywords:** osteoporosis, anti-resorptive drugs, therapeutic adherence, bone mineral density, biochemical markers, serum parathyroid hormone, calcium, alkaline phosphatase, 25-hydroxy vitamin D

## Abstract

Denosumab is a human monoclonal antibody that neutralizes RANKL, a cytokine able to interact with the RANK receptor on preosteoclasts and osteoclasts, decreasing their recruitment and differentiation, leading to a decreased bone resorption. The aim of this observational real-life study was to analyze adherence to denosumab therapy and assess its efficacy in increasing bone mineral density (BMD) and modulating biochemical skeletal markers following previous treatments with bisphosphonates in a group of post-menopausal women with osteoporosis. Women were recruited in the specialized center from March 2012 to September 2019. Biochemical markers were recorded at baseline and every six months prior to subsequent drug injection. Dual X-ray absorptiometry was requested at baseline and after 18/24 months. Comparing BMD at baseline and after denosumab therapy in naive patients and in those previously treated with bisphosphonates, a positive therapeutic effect was observed in both groups. The results of our real-life study demonstrate, as expected, that BMD values significantly increased upon denosumab treatment. Interestingly, denosumab showed an increased efficacy in patients previously treated with bisphosphonates. Moreover, biochemical markers data indicate that osteoporotic patients, without other concomitant unstable health conditions, could be evaluated once a year, decreasing the number of specialistic center access.

## 1. Introduction

During the last decades, in industrialized countries a significant increase of life expectancy with a gradual aging of the population has been observed. By 2060, America’s population of adults over the age of 65 is expected to double the current number [1]. In this scenario, the prolonged life expectancy, the inappropriate eating behaviors and lifestyle have contributed to the increased incidence of osteoporosis, a skeletal metabolic chronic disease characterized by compromised bone strength due to the reduction of both bone quantity and quality. The decreased bone strength leads the affected subjects to an increased risk of developing vertebral and non-vertebral fragility fractures, with increased morbidity, mortality, health service utilization, health care costs, and deterioration of quality of life [2,3,4,5,6,7,8].

All pharmacological treatments proposed in the last decades for osteoporotic patients aimed to reduce the risk of osteoporosis-related fractures by increasing bone mineral density and improving bone quality by modulating bone remodeling as demonstrated in controlled randomized clinical trial [9,10,11,12,13]. 

Denosumab, one of the latest pharmacological treatment developed as anti-osteoporotic treatment, is a fully human monoclonal antibody with high specificity to human RANKL, a receptor activator of NFκB ligand that is an essential factor in activating osteoclast differentiation and activity and, thus, activating bone resorption. This antiresorptive drug is administered by a subcutaneous injection every 6 months and it reversibly reduces osteoclast number and activity decreasing bone resorption. On the basis of this mechanism, denosumab has been demonstrated to significantly decrease new vertebral, nonvertebral, and hip fractures for up to 10 years of treatment in post-menopausal women affected by osteoporosis as demonstrated by the Fracture REduction Evaluation of Denosumab in Osteoporosis every 6 Months (FREEDOM) Extension trial [14,15,16]. 

Few studies, however, have evaluated adherence and efficacy of denosumab treatment in increasing BMD and potentially modulating biochemical skeletal markers in everyday life outpatients [17,18,19]. Thus, the aim of this observational real-life study was to analyze adherence to denosumab therapy and assess its efficacy in increasing bone mineral density (BMD) and modulating biochemical skeletal markers following previous treatments with bisphosphonates in a group of post-menopausal women with osteoporosis. In particular, it was evaluated and compared BMD at baseline and after denosumab therapy in either naive patients or patients previously treated with anti-resorptive drugs, since some previous studies suggested potential altered response to denosumab upon previous antiresorptive drugs [20,21,22]. 

## 2. Materials and Methods

### 2.1. Recruitment

Four hundred twenty-eight post-menopausal women (mean age = 71.7 ± 8.6 years) were recruited for this study. All subjects were affected by osteoporosis and referring to the specialized center for the management of osteoporosis in Policlinico Umberto I Hospital, Rome, from March 2012 to September 2019. Patients were evaluated every six months from the date of the initial prescription. The study respects the Declaration of Helsinki and was approved by the local Ethical Committees of the institution (prot 648/17) involved and met the guidelines of the responsible governmental agency.

### 2.2. Inclusion Criteria

Inclusion criteria were fixed on the basis of a previous work and AIFA’s advices for drug treatments: diagnosis of post-menopausal osteoporosis, possibility to receive a prescription during the study period according to the Italian reimbursement criteria for anti-osteoporotic pharmacological treatment [23,24].

Specifically, subjects with: (i) at least one prevalent hip or vertebral fracture (with contraindications to take oral pharmacological treatment), (ii) subjects aged ≥50 years with a femoral neck (FN) T-score ≤ −4 standard deviations (SD), (iii) subjects aged ≥50 years with a FN T-score ≤ −3 SD and at least one risk factor (family history of fragility vertebral or hip fractures, rheumatoid arthritis and other connective tissue diseases, previous wrist fragility fracture, early menopause, chronic corticosteroid therapy) were included.

The study was approved by the local University Institutional Review Board (prot n5061/18) and all participants gave their written informed consent before study participation. 

### 2.3. Intervention

Before starting the new pharmacological treatment, the referring specialist or a resident fellow explained to the patients the mode of action and the method of taking the drug. Treatment consisted of a subcutaneous injection of denosumab (Prolia^®^, Amgen Inc. Thousand Oaks, CA 91320-1799, USA) 60 mg/every 6 months. Each patient was clinically evaluated before the first administration of denosumab (baseline) and then every 6 months, before the following injection, until the end of the observational study period. All patients continued taking all other concomitant therapies, including calcium and/or vitamin D supplementation. The specialist allowed all patients, or their relatives, to contact the clinical group via e-mail or telephone in case of doubts, questions or important communications. At the date of the initial prescription (baseline) the following data and information (y/n) were collected: age (years), age of menopause (years), weight (kg), height (m), body mass index (BMI), history of smoking; family history for osteoporosis; family history for fragility fractures; history for fragility vertebral and/or femoral and/or any other skeletal site fractures; previous hormonal replacement therapy (HRT), and/or chronic corticosteroid therapy; any previous and/or current supplementation with calcium and/or vitamin D and any previous antiresorptive therapy with bisphosphonates (BPs, including risedronate, alendronate, clodronate, zoledronic acid, ibandronate) or strontium ranelate or teriparatide. BMD [T-score at lumbar spine (LS), total proximal femur (F) and femoral neck (FN)] was evaluated by dual X-ray absorptiometry (DXA) measurement. In regards of the biochemical markers, serum parathyroid hormone (PTH), calcium (Ca), alkaline phosphatase (ALP), and 25(OH) vitamin D3 (VitD) were analyzed. Since this is a real-life study, blood tests were not centralized in our hospital center, but patients performed them in the morning between 07.00 a.m. and 09.00 a.m. after an overnight fast in other clinical centers which are required to use standardized methods (Table 1). As previously mentioned, some patients were under medications for other diseases, which however did not affect bone metabolism.

Weight (kg) and height (m) as anthropometric parameters were recorded to calculate BMI, according to the following formula: BMI = weight/height^2^ (kg/m^2^). 

Osteoporosis was diagnosticated by DXA, based on the T-score for BMD less than or equal to −2.5 SD [25]. At baseline visit, lumbar, total proximal and femoral neck DXA were requested. Vertebrae affected by fractures, or other structural change or artifact, were excluded following the International Society for Clinical Densitometry—ISCD —official guidelines. Left hips affected by fractures, and underwent hip replacement surgery, were excluded and substituted by measurement of right hips. The following DXA was required after 18/24 months. 

Biochemical markers were measured at baseline, approximately a week before drug administration, and at 6, 12, 18 and 24 months of denosumab treatment, in order to allow the patients to bring results to the visit.

For all subjects, bone density at lumbar spine, total proximal femur and femoral neck (LS T-score, F T-score and FN T-score) at baseline, were compared with values collected over time.

Of all participants who were prescribed denosumab, 43 dropped out (10.04%) before the end of the study. Reasons for drop-out are reported in the results section. 

Since denosumab is also prescribed in post-menopausal women with hormone-responsive breast cancer under aromatase inhibitors (AI) adjuvant therapy, denosumab was prescribed to 53 women affected by breast cancer (BC group) [26]. In fact, as estrogens stimulate breast cancer cells, AI are aimed to inhibit the production of endogenous estrogens from aromatase in adipose tissue. On the other hand, low levels of estrogens can induce bone loss predisposing to osteoporosis and fractures [27,28,29]. BC group was excluded from the study for reasons that are reported in the results.

### 2.4. Statistical Analysis

Statistical analysis was performed on the following parameters: age, weight, height, BMI, age of menopause, BMD (LS T-score, F T-score, FN T-score), PTH, ALP, Ca and VitD. Differences between the means of above cited parameters were estimated using a two-sample t-test comparison between women with and without breast cancer. Pearson’s correlations among above cited parameters were estimated. The chi-square test was used to compare variables’ frequencies among groups (normal BMD, osteopenia, osteoporosis).

ANOVA_RM (repeated measures) were performed on BMD, PTH, ALP, Ca and VitD for different times. The time of the measurements were called T0 (baseline) and T1, T2, T3, T4, T5, T6, T7, T8 and T9 corresponding to 6, 12, 18, 24, 30, 36, 42, 48, 54 months, respectively. Statistical significance was defined as *p* ≤ 0.05. All statistical analysis was performed by SPSS version 24.0 software (SPSS Inc., Chicago, IL, USA).

## 3. Results

As previously described in the material and methods section, data were collected for each patient for a maximum of 8 years every six months from the beginning of the therapy, individually. Thus, we estimated and reported adherence of each patient, drop out and reason of it, densitometric analysis, biochemical markers, eventual missing data. To increase the power of the analysis, we considered the first 24 months of therapy for each patient in order to have a satisfactory number of patients to be considered. 

### 3.1. Drop-Out

Of 428 enrolled subjects, 43 dropped out (10.04%) before the end of the study: in particular, the majority dropped within the first 12 months of therapy (87%), whereas the rest (13%) dropped out after 12 months. Thus, from our data it results that outpatients had an adherence to therapy of 89.6% up to the end of therapy. As depicted in Table 2, reasons of drop out were: death (6/43, 14%), refusal of therapy (13/43, 30.2%), unspecific disturbs (2/43, 4.6%), fear of collateral effects (4/43, 9.4%), fear of injective drugs (3/43, 6.9%), suggested by another specialist to change therapy (10/43, 23.2%), followed in another specialized center (5/43, 11.6%) (Table 2). Indeed, several studies have demonstrated that higher number of drop out occurs within the first 12 months of chronic pharmacological intervention [30,31,32]. 

### 3.2. Comparison between Groups of Women with or without Breast Cancer

Two-sample t-test comparison between the group of 53 women with breast cancer (BC) and the group of 332 women without breast cancer (NoBC) depicted significant differences in age (*p* < 0.01). Consequently, the two groups could not be studied as a single population and BC women in therapy con denosumab were not included in the statistical analysis of this study to address the aim of the study. 

### 3.3. Partition between Naive Subjects and Subjects Treated with Previous Therapies

Of the 332 remaining patients (mean age = 72.8 ± 7.9 years), 237 women had at least one vertebral fragility fracture and 16 women had a previous femoral fracture.

From this group of 332 women, 126 subjects were considered naive (no previous anti-osteoporotic therapy), 122 were previously treated with BPs therapy, 43 either teriparatide or strontium ranelate and 41 had received two or more of the above-mentioned therapies, as shown in Table 3. In order to compare naive group (NG) with BPs previously treated group (BPG) data, the 84 subjects who received teriparatide or strontium ranelate prior to denosumab were not included in statistical analysis (ANOVA).

### 3.4. Pearson’s Correlations at Baseline

Pearson’s correlations among main parameters at baseline were estimated.

Interestingly, analysis showed that BMD femoral neck (FN T-score) was significantly negatively related with age (*p* < 0.01). Correlation values showed that menopause age was positively correlated with F T-score (*p* < 0.01) and FN T-score (*p* < 0.01), age was positively correlated with BMI (*p* < 0.01) and LS T-score (*p* < 0.05). VitD was inversely correlated with weight (*p* < 0.05) and Ca was negatively correlated only with PTH (*p* < 0.01). FN T-score was positively correlated with all parameters analyzed (*p* < 0.01) except Ca and VitD, while it was negatively correlated with PTH (*p* < 0.05).

### 3.5. Efficacy of Denosumab Therapy in Increasing BMD

We compared BMD values at lumbar spine, femoral and hip neck before (at baseline (T0) and after 24 (T4), 48 (T8) and 54 (T9) months of pharmacological treatment. In Figure 1, as expected, compared to baseline, BMD values significantly increased after 24 months of treatment. The chi-square test revealed that there was a significant change in BMD in patients in therapy with denosumab: evaluation showed that the number of subjects affected by a low BMD, compatible with an osteoporotic T-score was reduced after 48 months of denosumab therapy, whereas the percent of subjects with osteopenia was increased.

### 3.6. Previous Therapy

The interest was to evaluate whether previous therapies could have influenced the efficacy of denosumab therapy. Two different groups were considered: a group of subjects (*n* = 122) previously treated at least with one of the mentioned different bisphosphonates (BPG) and a naive group of subjects (*n* = 126) that have no taken any previous anti-osteoporotic medication (NG). BMD measurements (LS T-score, F T-score F and FN T-score) before (at baseline) and after denosumab therapy were compared by ANOVA_RM (Table 4).

A significant improvement (*p* < 0.01) was observed after denosumab therapy (Figure 2A–C). Differences between the two groups (previous therapy effect, see last line of Table 4) was not observed in Lumbar spine (Figure 2A) and Femoral (LS T-score, *p* = 0.91 and F T-score, *p* = 0.07) (Figure 2B) but was observed in Femoral Neck (FN T-score, *p* = 0.05) (Figure 2C).

### 3.7. Biochemical Markers

In order to study whether changes in biochemical marker values occurred during denosumab treatment in related variables, ANOVA (by ANOVA_RM) analysis were performed on PTH, ALP, Ca and VitD for different times. 

It was observed that PTH levels significantly increased from T0 to T1 (N = 36, *p* < 0.001) but did not change substantially until 24 months. As expected by an anti-resorptive drug, ALP levels markedly decreased following treatment (N = 51, *p* < 0.001). In later time-points, biochemical markers data were available for a lower number of patients, which however demonstrated no further changes as compared to those that occurred within the first 24 months of therapy. Indeed, serum calcium levels remained unchanged upon treatment (ANOVA, N = 39, *p* > 0.05) while serum VitD levels significantly increased from T0 to T1 but did not change significantly after T3 (ANOVA, N = 79, *p* < 0.001) (data not showed).

## 4. Discussion

This Italian observational real-life study evaluated the effect of denosumab in a group of post-menopausal women. The results showed a good response in BMD after 2 years of denosumab treatment in all skeletal sites evaluated, as demonstrated by the decreased percent of BMD values compatible with osteoporosis and the increased percent of BMD values compatible with osteopenia. As expected, these data, regarding the improvements of BMD in a population of osteoporotic out-patients, confirm results already showed in randomized clinical trial [33,34,35].

Moreover, our findings, comparing BMD at baseline and over time in naive group (NG) with BPs previously treated group (BPG), showed a significant improvement of BMD in all skeletal sites evaluated (*p* < 0.01) after denosumab treatment in both groups of patients, suggesting that previous BP therapy does not lead to a decreased efficacy of denosumab. On the other hand, it is interesting to note that the increase of BMD at femoral neck was significantly higher (*p* = 0.05) in patients previously treated with BPs as previously showed by Sánchez et al., suggesting a therapy effect [17]. These data could be explained by the different mechanism of action of these anti-resorptive drugs: while the effect of denosumab on bone metabolism might stop upon discontinuation, BPs might maintain the antiresorptive effect even after discontinuation [36].

Pearson’s correlations at baseline among BMD regions confirm FN T-score as the most reliable measure, being negatively and positively correlated with age and menopause age, respectively. Indeed, our data further indicate that older patients have a lower FN T-score at baseline and that, as expected, an older age at menopause has a protective effect on the bone, likely due to circulating estrogens, leading to higher FN T-score. Further, considering other parameters, VitD was inversely correlated with body weight, as calculated by BMI, since in the present study body composition data were collected. Nevertheless, on the base of BMI of the patients, we can speculate that higher levels of fat mass could contribute to low VitD levels, since numerous studies, including the ones of our group, have shown that obese/overweight subjects have lower vitamin D levels, considering that the storage of this fat-soluble vitamin is mainly in the adipose tissue [37,38,39,40].

Interestingly, the measurement of bone biochemical markers after the first infusion of denosumab, showed a significant increase of serum PTH and VitD levels after 6 months of denosumab treatment (*p* < 0.001), confirming previous observation by Makras and colleagues [41]. But, unlike described in a more recent manuscript, PTH levels did not further change after the first six months of treatment [42]. The increase of circulating PTH after 6 months of treatment was not associated with lower levels of VitD and calcium, but this rise could be caused indirectly as result of an increase in bone anabolic activity in response to the strong decrease of osteoclasts (OCs) activity, as hypothesized by Nakamura et al. [43].

In regard of the analysis of ALP levels, we only considered the first 12 months of treatment during which, generally, bone formation markers are lowered upon anti-resorptive therapy [44]. 

Unfortunately, a lower number of patients had blood test analysis data, but in those patients, who have been followed for a longer period, no further changes were observed compared to those that occurred within the first 24 months of therapy.

The increase of VitD levels after 6 months is likely due to the supplementation leading to an optimization. In fact, only a small proportion of patients (38.8%) used to take supplementation of VitD before starting denosumab therapy, but most of them did not assume it regularly and the others did not take any VitD supplementation at all.

It is important to enlighten that, as previously explained, this is a real-life study and not a randomized clinical trial, thus data were collected for 8 years every six months, but for several different reasons (compliance of patients to prescription, economic reason) not all subjects could always afford biochemical tests. Although this might be a limitation, we have observed and reported this lack of data, which might often occur in a real-life approach of patients affected by osteoporosis. Nevertheless, BMD results confirmed that the first 24 months are relevant to test the efficacy of denosumab therapy. Moreover, our results show that denosumab therapy can influence PTH, ALP, VitD and Ca levels within the first 24 months of treatment, while no further significant modification can be observed after then assuming that, once stabilized, it is not essential to request blood tests every six months. Indeed, the outcomes of this study might be very important from a socio-economic point of view and suggest that, stable patients, without other concomitant unstable health condition, can be evaluated after 12 months and the blood tests control at the same time. In everyday life this clinical approach could be most useful and in line with the purpose of saving social and health costs, but also maintaining a good quality of life of these fragile subjects.

The study has some limitations. Due to the nature of the study perform on outpatients, we could not have centralized blood test evaluations, which would have allowed a more accurate analysis of all the bone markers, nevertheless, the examination of biochemical markers can be considered comparable within the limitations of a real life study. In addition, the number of patients in the longer therapy period are a smaller number as compared with the more recent enrolled patients. 

## 5. Conclusions

The dropout was confirmed within 12 months. Stable patients, without other concomitant unstable health condition, can be evaluated after 12 months. Our results indicate that denosumab increases his efficacy in patient previously treated with bisphosphonates.

## Figures and Tables

**Figure 1 ijerph-18-01728-f001:**
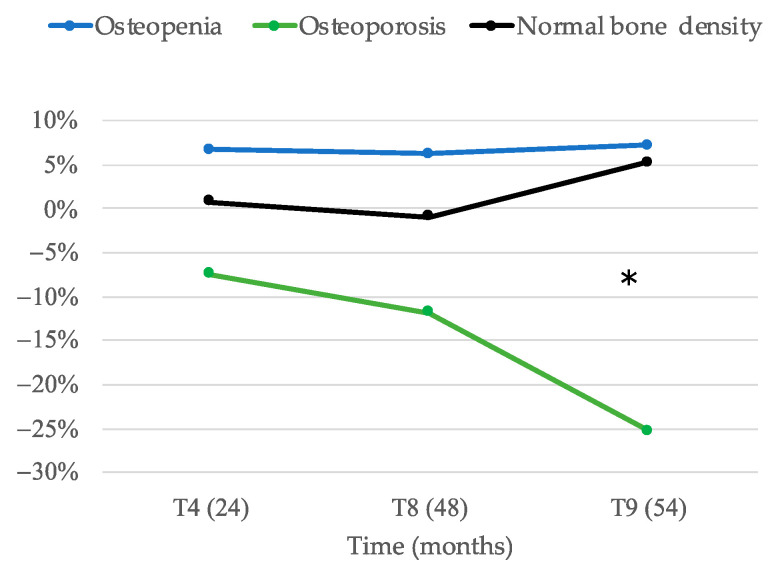
Percentage of change as compared to baseline of subjects with osteoporosis, osteopenia and normal bone density during treatment (* *p* < 0.05).

**Figure 2 ijerph-18-01728-f002:**
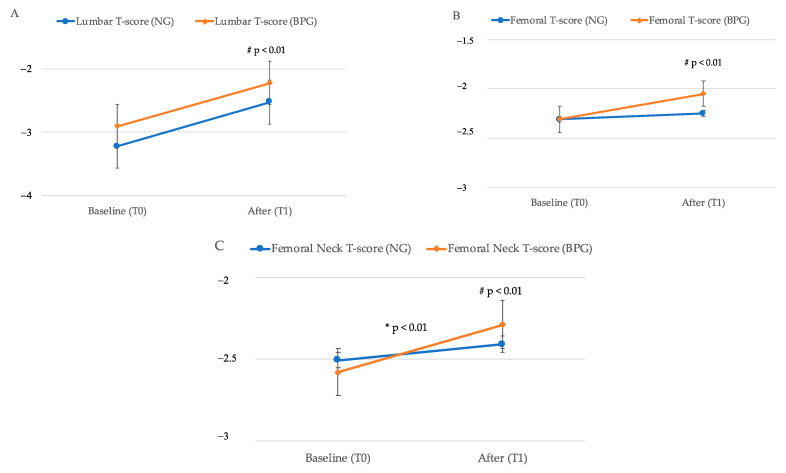
Comparison between BMD T-score (mean value, standard error) in NG (blue line) and BPG (red line) group at baseline T0 and after T1 denosumab. (**A**) Lumbar spine T-score measurement; (**B**) Femoral T-score measurement; (**C**), Femoral Neck T-score measurement. * Represents statistical significant difference effect for time (*p* ≤ 0.05), while ^#^ indicates statistical significant difference effect for time therapy (*p* ≤ 0.05) as more extensively reported in Table 4.

**Table 1 ijerph-18-01728-t001:** Study population characteristics at baseline (before denosumab treatment).

Parameters	Value
Age (years)	72.8 ± 7.9
Age of menopause (years)	48.7 ± 4.7
BMI (kg/m^2^)	24.8 ± 4.5
Habitual smokers (n, %)	21 (6.3%)
Family History of osteoporosis (n, %)	107 (32.3%)
Family History of fragility fractures (n, %)	47 (14.1%)
At least 1 vertebral fragility fracture (n, %)	237 (71.3%)
FN osteoporotic fracture (n, %)	16 (4.8%)
Osteoporotic fracture at any other site (n, %)	33 (9.9%)
Previous HRT (n, %)	32 (9.6%)
History of glucocorticoid use (n, %)	3 (0.9%)
Previous supplementation with Ca/VitD (n, %)	129 (38.8%)
Prior anti-osteoporotic treatment (n, %)	206 (62.0%)
LS T-score (SD)	−2.7 ± 1.2
F T-score (SD)	−2.1 ± 0.9
FN T-score (SD)	−2.5 ± 0.7
Serum PTH (pg/mL)	50.6 ± 23.3
Serum calcium (mg/dL)	9.5 ± 0.6
Serum ALP (U/L)	94.5 ± 52
Serum 25(OH)vitamin D_3_ (ng/mL)	32.8 ± 15

**Table 2 ijerph-18-01728-t002:** Causes of dropouts (*n* = 43/428, 10.04%).

Causes	(*n*, %)
Death	6 (14.0%)
Refusal of therapy	13 (30.2%)
Unspecified disturbances	2 (4.6%)
Fear of collateral effects	4 (9.4%)
Fear of injectable drugs	3 (6.9%)
Suggested to change therapy ^1^	10 (23.2%)
Followed in another center	5 (11.6%)

^1^ by another specialist.

**Table 3 ijerph-18-01728-t003:** Prior anti-osteoporotic treatment (*n* = 206/332, 62.04%).

Treatment	(*n*, %)
Bisphosphonates	122 (59.2%)
Strontium ranelate or Teriparatide	43 (20.9%)
Multiple treatment	41 (19.9%)

**Table 4 ijerph-18-01728-t004:** BMD comparison between NG and BPG group.

	Group	LS T-Score	F T-Score	FN T-Score
Number of subjects	NG (n)	29	28	30
	BPG (n)	37	37	35
ANOVA results	Time effect (p) ^1^	<0.01	<0.01	<0.01
	Time × therapy (p) ^2^	0.91	0.07	0.05

^1^ Efficacy of denosumab; ^2^ differences in efficacy of denosumab between the groups; n, number of subjects; p, statistical probability.

## Data Availability

The data presented in this study are available on request from the corresponding authors.

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
