# Peer review of "Efficacy of Denosumab Therapy Following Treatment with Bisphosphonates in Women with Osteoporosis: A Cohort Study"

_ijerph, 2021, doi:10.3390/ijerph18041728_

Round 1
Reviewer 1 Report
Manuscript ID: ijerph-1063907
Effectiveness of denosumab therapy after treatment with 2 bisphosphonates in women affected by osteoporosis: a cohort 3 study
Overview: The aim of this observational study was to analyze adherence to denosumab therapy and assess its efficacy in increasing bone mineral density (BMD) and modulating biochemical markers following previous treatments with bisphosphonates in a group of post-menopausal women with osteoporosis. They found that BMD significantly increased following denosumab treatment. Denosumab showed increased efficacy in patients previously treated with bisphosphonates. The study is not a randomised controlled trial but it is a real life observational study and it is clearly labelled as such. There are many grammatical and spelling errors throughout the manuscript, including the figures and the labels of the graphs. These errors must be corrected by the authors.
In summary, this is a nice study that should be of interest to the readership of this journal. A number of revisions would improve the flow and readability of the manuscript.
Major comments:
- The title should be revised as follows: Efficacy of denosumab therapy following treatment with bisphosphonates in women with osteoporosis: a cohort study
- Affiliations: the authors should either use English or Italian in the affiliations. For example, if they are going to use English for the affiliations, Roma should be changed to Rome.
- The aim of the study should be revised as follows: The aim of this observational real-life study was to analyze adherence to denosumab therapy and assess its efficacy in increasing bone mineral density (BMD) and modulating biochemical skeletal markers following previous treatments with bisphosphonates in a group of post-menopausal women with osteoporosis.
- It might be better if the word effectiveness is simply replaced with efficacy throughout the manuscript.
- The keywords should include some of the biochemical markers that were measured.
- Introduction is well written.
- The methods and inclusion criteria are well described. Details of the intervention are sufficiently outlined.
- Results are clearly outlined but it is not clear if simply plotting the percentage of subjects against time is sufficient in figure 1. Changes compare to baseline might have been more appropriate. Statistics should be included in figure 2.
- The discussion is well written and detailed and so is the conclusion but the authors should include a section on their own perceived limitations of the study. The authors mention that this is not a randomised clinical trial but clearly this is an obvious shortcoming of the study.
In summary, this is a nice study that should be of interest to the readership of this journal. A number of revisions would improve the flow and readability of the manuscript. I would recommend that the authors revise the paper.
Author Response
We thank the Reviewer for her/his time and for the comments and suggestions that we believed have improved the manuscript. We have modified the manuscript accordingly and further addressed all the points raised by the Reviewers.
Major comments:
- The title should be revised as follows: Efficacy of denosumab therapy following treatment with bisphosphonates in women with osteoporosis: a cohort study
We thank the Reviewer for the useful advice. We have modified the title accordingly to his/her suggestion.
- Affiliations: the authors should either use English or Italian in the affiliations. For example, if they are going to use English for the affiliations, Roma should be changed to Rome.
We thank the Reviewer for having noticed this point. We have now corrected the affiliation in the revised version of the manuscript.
- The aim of the study should be revised as follows: The aim of this observational real-life study was to analyze adherence to denosumab therapy and assess its efficacy in increasing bone mineral density (BMD) and modulating biochemical skeletal markers following previous treatments with bisphosphonates in a group of post-menopausal women with osteoporosis.
We thank the Reviewer for the helpful suggestion. We have modified the sentence regarding the aim, and the modified sentence can be now found in both Abstract section at page 1, line 22-23 and Introduction section at page 2 row 100 of the revised manuscript.
- It might be better if the word effectiveness is simply replaced with efficacy throughout the manuscript.
We have replaced the word effectiveness with efficacy throughout the revised version of the manuscript.
- The keywords should include some of the biochemical markers that were measured.
We are grateful to the Reviewer for his/her useful suggestion. We have added, additional keywords including some of the biomarkers.
- Introduction is well written.
- The methods and inclusion criteria are well described. Details of the intervention are sufficiently outlined.
- Results are clearly outlined but it is not clear if simply plotting the percentage of subjects against time is sufficient in figure 1. Changes compare to baseline might have been more appropriate. Statistics should be included in figure 2.
Figure 1 was modified as requested by the Reviewer showing the changes as compared to baseline (T0). Significant differences have now been indicated. The chi-square test was used as explained at pag 4 line 154-155. Results are also further described at page 5, lines 211-213 of the revised version of the manuscript.
As suggested by the Reviewer, Figure2 was changed with a revised one (panel A, B and C) where appropriate statistics has been included. Figure legend has been modified accordingly.
- The discussion is well written and detailed and so is the conclusion, but the authors should include a section on their own perceived limitations of the study. The authors mention that this is not a randomized clinical trial but clearly this is an obvious shortcoming of the study.
We thank for the positive comments. We have now added a section regarding study limitation, at the end of the Discussion section.
In summary, this is a nice study that should be of interest to the readership of this journal. A number of revisions would improve the flow and readability of the manuscript. I would recommend that the authors revise the paper.
We thank the Reviewer again for the careful revision of our manuscript.
Reviewer 2 Report
The study is very interesting thereby including a lot of patient data. The inclusion and exclusion criterias are clearly stated and the ethical approval number is included in the manuscript. Please find some major comments below:
Introduction:
Page 2, line 46: exclude "e" and include "and"
Methods:
The authors do not mention how biochemical parameters were assessed. Please include it in the methods' section.
Results:
Page 6, Figure 1: Did you perform any statistics for Figure 1? Obviously, there is a decrease in osteoporosis and since the figure lacks standards deviations, I kindly ask the authors to include any statement on statistical outcome.
Page 8, lines 276-311: there are no results on this discussion part, please include it.
Author Response
We thank the Reviewer for finding our manuscript interesting. We thank for the careful revision of our manuscript that we believed has improved upon the modifications suggested. We have revised the manuscript accordingly and addressed all point raised.
Introduction:
Page 2, line 46: exclude "e" and include "and"
We thank the Reviewer for having noticed the mistake. We have corrected it.
Methods:
The authors do not mention how biochemical parameters were assessed. Please include it in the methods' section.
We thank the Reviewer for his/her comment. Since this is a real-life study, biochemical analyses were not centralized in our hospital center. Patients performed biochemical tests in different laboratories, which usually use standardized method. We have reported this point in page 3, line 119 of the revised version of the manuscript. We have also included this point as a limitation of our study at the end of Discussion Section.
Results:
Page 6, Figure 1: Did you perform any statistics for Figure 1? Obviously, there is a decrease in osteoporosis and since the figure lacks standards deviations, I kindly ask the authors to include any statement on statistical outcome.
We thank the Reviewer for his/her comment. We have performed statistical analysis as described at page 4 from line 151 to line 161 of the revised version of the manuscript. Further discussion on the statistic outcomes and clinical significance is present in both the Results and the Discussion section. In addition, in regard of figure 1, we have added statistical difference within the graph in the modified version of the manuscript. Figure 1 was modified as requested by the Reviewer 1 showing the changes as compared to baseline (T0). Significant differences have now been indicated. The chi-square test was used as explained at pag 4 line 154-155. Results are also further described at page 5, lines 211-213 of the revised version of the manuscript.
Page 8, lines 276-311: there are no results on this discussion part, please include it.
As suggested by the Reviewer, we have described the results obtained in the study from line 276 to line 284 in the Results section. We have further discussed the results obtained in the study in the discussion section at page 9 from line 279 to 334.
Round 2
Reviewer 2 Report
Thank you for revising the manuscript appropriately. However, I am still not convinced that the evaluation of biochemical markers is comparable. How can the authors guarantee that the examination of biochemical markers is comparable and combinable (all patients were fasted or fed); patients were under any kind of medication; time point of testing is comparable (when did the patient undergo blood examination); etc.?
Author Response
We thank the Reviewer for his/her further revision of our manuscript. We have modified the manuscript accordingly to the comment and highlighted the revision in yellow/green.
Specifically, at row 121 and row 325.
